# Experiential Learning: Using Partition-Tree Weighting and MAML in the Continual and Online Setting

## Abstract

Learning from experience means remaining adaptive and responsive to errors over time. However, gradient-based deep learning can fail dramatically in the continual, online setting. In this work we address this shortcoming by combining two meta-learning methods: the purely online Partition Tree Weighting (PTW) mixture-of-experts algorithm, and a novel variant of the Model-Agnostic Meta-Learning (MAML) initialization-learning procedure. We demonstrate our approach, Replay-MAML PTW, in a piecewise stationary classification task. In this continual, online setting, Replay-MAML PTW matches and even outperforms an augmented learner that is allowed to pre-train offline and is given explicit notification when the task changes. Replay-MAML PTW thus provides a base learner with the benefits of offline training, explicit task sampling, and boundary notification, all for a $O(\log_2(t))$ increase in computation and memory. This makes deep learning more viable for fully online, task-agnostic continual learning, which is at the heart of general intelligence.

## 1 Experiential Learning

Learning is by definition a process. Yet machine learning models are typically evaluated and deployed as the static, learned product of training rather than as systems that can continually adapt and learn. This train-and-deploy approach has created successful models when there is sufficient appropriate data. But even appropriate data ages and the mismatch between training and deployment only grows. Truly robust and general machine learning systems incorporate continual adaptation and improvement as the world changes around them.

In this paper we refer to this learning challenge as *experiential learning*: all data available to the learning system is sequential, evaluation and learning are simultaneous and ongoing, and there are no guarantees that the underlying data distribution remains stationary. These characteristics all occur to varying degrees in other machine learning frameworks, and both *continual learning* and *online learning* may cover some or all aspects. Yet continual learning in particular remains heavily influenced by the historic development of supervised learning, where the task is defined by a static dataset. An experiential learning problem is not only sequential and potentially changing over time, it is defined as a process and cannot be summarized or evaluated with a static dataset.

To address the experiential learning challenge we propose a new meta-learning algorithm that combines the strengths of two existing algorithms. Partition-Tree Weighting (PTW) provides the basis for learning from continually changing data sources (Veness et al., 2013). Model-Agnostic Meta-Learning (MAML) provides a mechanism for learning good initial network weights (Finn et al., 2017), which PTW is able to use to improve ongoing performance. A simple extension to MAML, which we call Replay-MAML, removes MAML's reliance on task-specific training. Combining these two techniques gives us Replay-MAML PTW, which requires no explicit definition of tasks and no signal of task changes. It learns from sequential data, incrementally updating its model and adjusting based on iterative feedback rather than requiring explicit re-training. Replay-MAML PTW can match the performance of the best task-aware counterparts, while being itself a fully experiential learner.

## 2 Background: from product to process

An assumption underlying many machine learning techniques is that data is independent and identically distributed (i.i.d.): the distribution of features ($\mathbf{x}$) and the conditional distribution of the correct class ($y|\mathbf{x}$) are stationary. In this regime, the performance of the model is often measured with *offline testing*, where a subset of the data not used during training is used to evaluate model performance. However, the real world — along with many interesting applications in it — is non-stationary: the observed data distribution can change over time. Even the task definition, *i.e.*, the conditional probability of the correct class, can shift.

*Continual learning*, or machine learning from sequential and non-stationary data (Hadsell et al., 2020; Aljundi, 2019), has become increasingly recognized as an important area of study. Yet the conception of learning as discovering a fixed mapping continues to drive much of this research, with evaluation defined separately from the agent's stream of experience. Many continual learning techniques rely on pretraining with a separate, offline data source. And even *online continual learning* tends to signal incremental training or fine-tuning, with evaluation still be understood in terms of performance on external tasks. *Online learning* is the study of learning from streaming data, where the model is trained iteratively and evaluated on every new input (Shalev-Shwartz, 2012). It overlaps strongly with our experiential learning framework. But formal definitions allow for the incorporation of additional (*i.e.*, non-experiential) knowledge (Shalev-Shwartz, 2012). In reinforcement learning, the term 'online' may refer to incremental learning, in contrast to batch learning, even when the environment itself is static(Sutton & Barto, 2018).

Similarly *reinforcement learning* has traditionally been defined in terms of an agent's continual interaction with an environment (Sutton & Barto, 2018). In spite of this, reinforcement learning systems are often evaluated offline, via the application of a learned (but now static) value function or policy in a newly-sampled environment (Sutton et al., 2007). *Lifelong learning* is frequently used to refer to systems that must continually improve or transfer knowledge from past to future tasks (Thrun, 1994), but framing generalization as *cross-task transfer* tends to bring back the emphasis on offline evaluation with a fixed set of tasks.

### 2.1 The Experiential Learning Framework

Taking inspiration from reinforcement learning and its agent-environment framework, *experiential learning* emphasizes the interactive nature of a learning system operating in the world, bounded by time and available inputs. The data for an experiential learning problem is inherently sequential and may be non-stationary. In fact, sequential data is the only data there is. As an i.i.d. learning problem is defined by a dataset, so an experiential learning problem is defined by the sequence of data. This does not preclude the existence of distinct tasks (sequences of data that can be accurately modeled as an i.i.d. batch), nor does it necessarily limit an agent from creating and using task-based models. External tasks, offline evaluation, and counter-factual experiments may be useful for understanding the behaviour and limitations of a learning agent, but not for evaluating performance. The experiential learning problem is defined by the process.

In other words, experiential learning is:

- **Embedded:** The data has a temporal context. Past experience and its context are lost unless actively stored. Future data is uncertain.

- **Immediate:** Data is only accessible through interaction and always demands a response. Evaluation is constant but only in terms of the data experienced.

- **Ongoing:** There is no predefined end. The data distribution may change, remain stationary, or even recur, but experience continues and the agent must react appropriately.

### 2.2 Related work

In supervised learning, *continual learning* describes learning from sequential and non-stationary data (Hadsell et al., 2020; Aljundi, 2019). The most common continual learning framing is to conceptualize the learning problem as sequence of i.i.d. tasks: a Markov chain or piecewise-stationary sequence, where the distribution

of tasks is itself stationary, known, and often explicitly identified. Learning may proceed as in the stationary setting for the duration of the tasks, or regularization might be used to constrain the learning process in order to resist changing weights that succeeded in past tasks. Many continual learning approaches, while allowing for changing tasks, require that tasks be explicitly identified or presented in a particular order, as in task-incremental learning (Lange et al., 2023). Most assume the existence of an underlying structure that can simultaneously represent all tasks, and only a few *task-agnostic* learning systems allow for learning without task changes being signalled explicitly (Caccia et al., 2020; He et al., 2019; Kirkpatrick et al., 2017). Even in these cases, successful learning is typically described in terms of offline performance on a predetermined set of tasks.

Continual learning is often considered synonymous with the study of *catastrophic forgetting*, which is when a learning agent that has previously performed well on a task thoroughly fails, because of intermediate training on a related but different task (Aljundi, 2019). Importantly, this can happen even when the deep learning network is in principle capable of representing every task (Lange et al., 2023; Hadsell et al., 2020). There is some evidence this 'forgetting' occurs even within the same task (Lange et al., 2023; Fedus et al., 2020). The balance between maintaining performance while adapting as necessary is known as the *plasticity/stability* trade off. Clearly such a trade-off applies to all learning systems, but it is usually studied in terms of explicit task definitions and held-out test performance. Even when explicitly studying the continual learning setting, results rarely investigate performance during learning and even then do so in terms of full evaluation of offline test sets (Lange et al., 2023).

Recent work in supervised learning by Dohare et al. (2022) and in reinforcement learning by Abbas et al. (2023) has shown that deep continual learning may have even more fundamental issues. *Catastrophic loss of plasticity* is the abrupt failure of adaptation in an online gradient-based deep learning networkDohare et al. (2022). This is not merely poor performance of the network weights on current, past or future tasks. It is a change in online learning behaviour from the what Sutton refers to as the 'alligator graph' (where the sequence of successful within-task learning creates a quickly-rising curve followed by abrupt) to In Dohare's work the behaviour was consistent across all network architectures, activation functions, and learning algorithms tested. Re-initializing the model completely can of course restore plasticity, but only by losing all learning. Dohare explored methods for partial reinitialization, and Abbas' work demonstrates that CReLu activation functions can alleviate the loss of plasticity. The exact cause and solution of the failure remains an active area of investigation. Although saturation, exploding weights, vanishing gradients, dead neurons and more play a part in the collapse, it appears that any deep continual learning system will eventually face this issue and both the cause and the solution remain unclear.

Through our exploration of experiential learning regimes we found similar catastrophic failures with standard gradient-based approaches across a range of tasks, network architectures, and optimisation criteria. Results pertinent to our main investigation will be presented in later sections, but a full investigation is beyond the scope of this paper. However, our definition of experiential learning is largely in response to this mystery, and we hope it proves to be a useful framework for future investigations of catastrophic loss of plasticity.

## 3 Experimental Testbed

Throughout this paper we use a modification of the standard MNIST benchmark (Deng, 2012), that we call Switching MNIST. Switching MNIST is explicitly constructed to provide an unending stream of tasks from a finite data source that we know can be well-represented by standard learning techniques. It is defined as an experiential learning problem but allows for offline testing and the incorporation of task knowledge, for comparison between experiential and non-experiential learners.

### 3.1 Switching MNIST

Switching MNIST maintains an internal task context to create a continual $k$-way classification problem from the MNIST dataset. The task context is defined by an input distribution and class mapping. On each timestep, Switching MNIST samples a batch of images from the current input distribution, scores the learner's accuracy against the current class mapping, and then provides the learner with the labeled batch.

It then may change the task context. In the results shown, tasks are resampled on each timestep according to a fixed *switch probability*, creating a piecewise-stationary, task-agnostic sequence[1]. The input distribution is uniform random over the subset of digits used in current class mapping[2]. To determine the class mapping for a new context, Switching MNIST uniformly samples $k * n$ labels from the 10 possibilities provided by MNIST, and assigns $n$ to each of the $k$ classes[3]. This creates a sequence with varying periods of stability, where the learner experiences both repeated and novel data distributions. Furthermore, over sufficiently long learning periods each label will appear in all classes equally often. Thus *catastrophic label interference*, or the complete reversal of previously-learned class mappings, is an unavoidable reality.

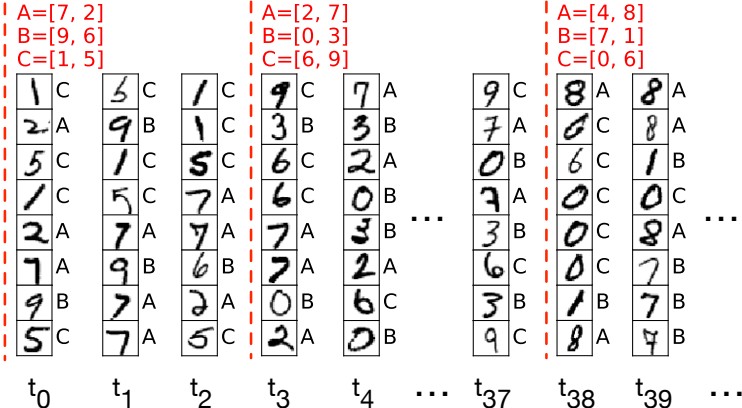

Figure 1: A sample sequence of experience from a Switching MNIST experience. Stochastic task switches are marked with a dashed red line. The current task is shown in red, though the agent sees only the image-class pairings and not the mappings from digit labels to classes. At each time-step, eight images are drawn from the current input distribution, then assigned a class according to the current mapping.

An example stream of experience from Switching MNIST is illustrated in Figure 1. The agent sees only the set of image-class pairings (shown in black). The environment information is shown in red, annotating the task-specific class mapping, with the dashed line indicating the beginning of a new task. The agent sees a random set of images from those involved in the current task, assigns a class to each, and its accuracy is recorded. The agent then updates its model. With some probability the environment may then switch the task. In general this is a fully task-agnostic setting: although the environment is piecewise stationary with discrete task changes, the agent is not given any indication of switches or prior knowledge of the underlying task structure.

Switching MNIST satisfies our experiential criteria while allowing for comparison with non-experiential learners. Labels may appear in any class, so learning must be embedded in time: the input-class mapping only applies within the current task context. Even the input distribution is embedded in the current context, as samples are drawn from only from the subset of digits currently assigned to a class. But because the environment is piecewise-stationary and constructs tasks with a defined distribution, it is possible to train non-experiential learners offline. This also allows us to provide the task-switch signal to non-experiential learners for direct comparison to their experiential counterparts[4]. Evaluation is in terms of per-step average accuracy on the current task, thus it is both immediate and constant. The random assignment of labels also allows us to turn the static MNIST dataset into an ongoing environment without being restricted by the size of the dataset. Using a simple classification setting, using two classes consisting of a single label, tasks are

---

[1]In existing literature 'task-agnostic' may mean only that task changes are not explicitly signalled. We use a stronger definition in this work where additionally the number and nature of tasks is not known.

[2]We use 'labels' or 'digits' when referring to the original MNIST label and 'class' when referring to class given by Switching MNIST's current context.

[3]We experimented with non-uniform class sizes and restricting label repetition on consecutive tasks, but these did not show interesting effects, so we restrict our presentation to the simpler case.

[4]Throughout this paper, we will use + in algorithm names to denote non-experiential counterparts to our basic learning algorithms: +Reset indicates the algorithm is notified when the environment changes context. ALG+ indicates pretraining was allowed.

quite likely to repeat. In more complicated settings, tasks may unlikely to repeat at all. Although the size of the dataset ensures that inputs will recur, their context is changing.

Results presented here specifically show performance for $k = 3$ with each class containing two randomly assigned labels. Each trial consists of one million timesteps, with a task remaining the same for 50 timesteps on average[5]. This creates a relatively difficult learning problem with very low probability of exact repeats[6]. Accuracy can range from 33.3% (uniform random) to a maximum of 98.6% (random on the first timestep of each task and perfect thereafter). Earlier experiments with two classes of a single label showed the same qualitative behaviour as presented here.

### 3.2 Continual Learning with Stochastic Gradient Descent

The underlying model for all learning agents in this paper is a standard stochastic gradient descent (SGD) learner that uses a simple yet sufficient neural network to minimize cross-entropy loss. The network has one convolutional layer (64 channels, 3x3 kernel, stride of 1) and 2 fully-connected linear layers (128 wide and 64 wide) with ReLu activation functions. Images are classified with a softmax over the output layer. With 10-class classification and simple SGD optimization, the network is complex enough to perfectly memorize the 60,000 image training set, but not all possible image-class assignments created in Switching MNIST. Furthermore, the network uses the number of classes from Switching MNIST, so is not able to classify all digits consistently.

Two versions of the SGD learner form the basis for our future comparisons, SGD and SGD+Reset[7]. In both cases the network weights are randomly initialized at the beginning of time and updated with one $\alpha$-weighted gradient step per timestep. The gradient is with respect to the cross-entropy loss, with $\alpha$ tuned independently for each experiment setting. We did not use momentum, gradient clipping, or normalization as these techniques hastened the onset of catastrophic loss of plasticity. SGD is initialized just once and updates continually thereafter, and so entirely matches the criteria for an experiential learner. It can transfer between tasks, but it also suffers from instability and loss of plasticity as we will show in Section 3.3. The task-aware version, SGD+Reset, receives an extra signal from the environment on timesteps when the task has changed. At such a point it randomly reinitializes the weights of its network. Although it cannot learn across task boundaries its performance provides a baseline for perfectly plastic learning.

### 3.3 SGD results in Switching MNIST

Figure 2 shows the initial performance of SGD and SGD+Reset averaged over 30 random seeds with shading indicating the 95% confidence interval[8]. SGD+Reset's average performance ($75.8 \pm 0.1\%$) over the first 100,000 steps is significantly higher than the uniform-random baseline (black dotted line at 33%), showing it learns quickly in spite of the short task length and random reset. In this setting SGD ($85.0 \pm 0.3\%$) quickly outperforms SGD+Reset. This indicates that the experiential learner is benefiting from cross-task transfer, in spite of the non-i.i.d. setting and catastrophic label interference.

However, if we extend the life of our continual learner, as shown in Figure 3, a surprising issue emerges. The accuracy of SGD+Reset naturally remains consistent, as reset makes its performance on each task independent. SGD has a dramatic collapse, and falls to accuracy of uniform random classification. This is not, as Figure 3a might suggest, due to any gradual decay in accuracy, but an abrupt and permanent plummet on each individual trial from peak to random performance. Within our 30 trials the earliest collapse happened after just 92,000 updates, and by one million updates all 30 learners had collapsed, as can be seen in Figure 3b. Our experiments confirm the findings discussed in Section 2.2: catastrophic loss of plasticity is a consistent phenomenon and surfaced in all settings we tried, if we just ran the online evaluation for long

---

[5]The switch probability was approximately 0.02 per timestep, as it was set at 0.0025 per sample with 8 samples per timestep.

[6]18,900 possible different tasks (210 sets of active labels with 90 possible class groupings)

[7]Recall the + signals non-experiential learning

[8]Unless otherwise specified all results are averaged over 30 trials and smoothed further by averaging over 500 bins across the timespan shown in each graph. As a result, graphs with longer horizons appear more smooth due to averaging over larger bin sizes. Shading always indicates the 95% confidence interval. The black dotted line, when present, indicates the uniform random baseline.

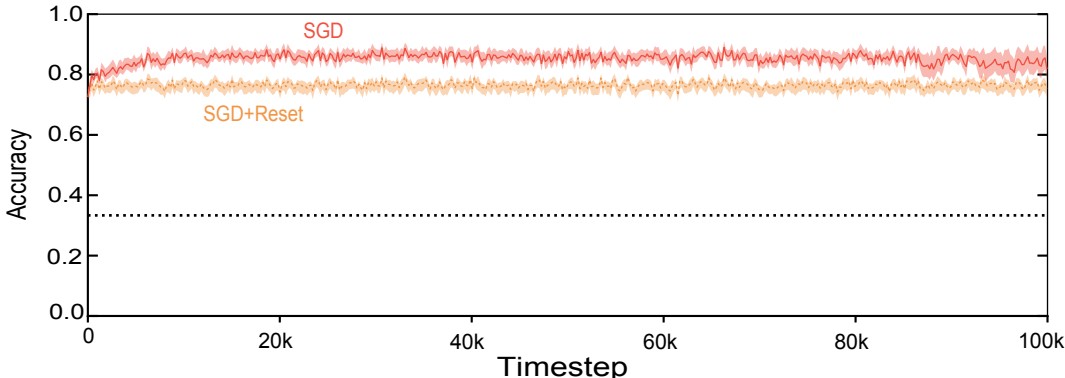

Figure 2: Accuracy of SGD and SGD+Reset over 100,000 timesteps, averaged over 30 random seeds. SGD is an experiential learner and consistently outperforms SGD+Reset, the task-aware counterpoint represented with a lighter dashed line. Both significantly improve on uniform random (the dotted black line).

enough. While SGD improves on the completely plastic SGD+Reset for sufficiently short experiments, the catastrophic loss of plasticity means SGD alone is not a solution to experiential learning.

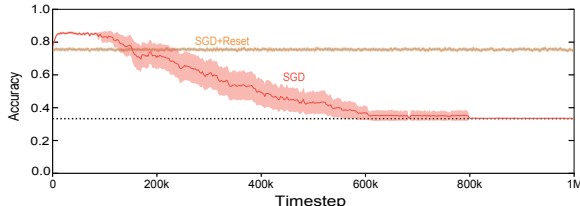

(a) Accuracy averaged over 30 trials. The performance of the experiential SGD learner, initially best, falls to uniform random.

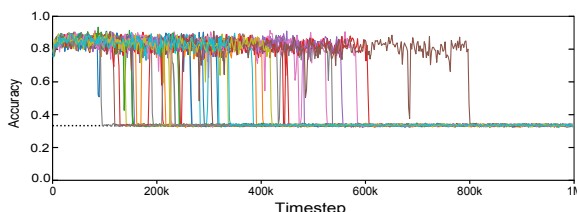

(b) In each of the 30 trials, the SGD learner eventually suffers a catastrophic loss of plasticity.

Figure 3: SGD accuracy over one million timesteps.

## 4 Partition-Tree Weighting

Partition-Tree Weighting (PTW) (Veness et al., 2013) provides a mechanism for generalizing a base learning algorithm to non-stationary problems, by maintaining copies of the base learner's models that are automatically reinitialized at set points in time. Using only $\lceil \log_2(t) \rceil$ models, PTW creates a mixture model that closely approximates the performance of the base learner on the best possible partition in hindsight (see Veness et al. (2013) for theoretical details). When learning across all experience is the best course of action, as in a typical i.i.d. learning task, PTW places most of its weight on the longest-running model. When unlearning is costly, as for some piecewise-stationary tasks, PTW places more weight on recently initialized models. It is able to do this without reference to explicit task definitions or boundaries.

### 4.1 PTW overview

PTW's internal structure is illustrated in Figure 4. The rounded black rectangles represent particular instantiations of the base learner. Each was initialized on the first timestep in the box and updated $2^d$ times according to their height in the tree. Although the tree must contain 31 different nodes to cover all possible binary partitions of the 16-timestep sequence, only the 5 that cover the current timestep can updated or used for prediction. Thus PTW maintains $d$ models and efficiently computes the cost of all partitions by

summarizing each subtree as it is completed. These summary statistics are represented in the figure by the blue boxes encompassing past subtrees.

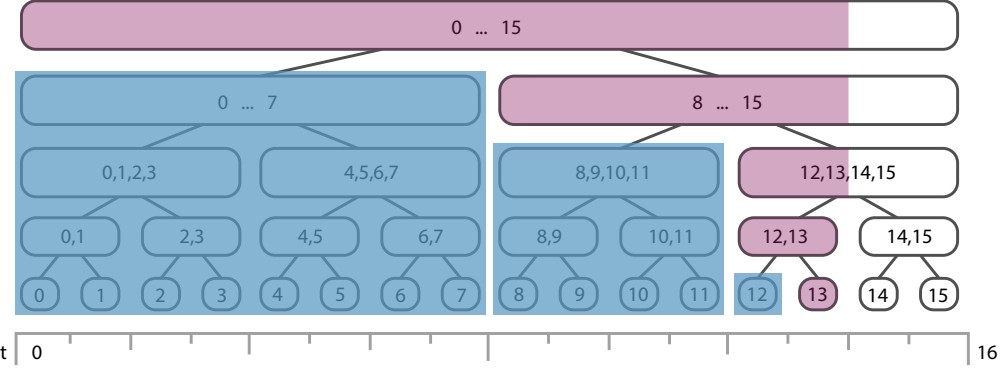

Figure 4: An illustration of PTW's internal structure. Each node (rounded black rectangle) in the binary tree represents one particular instantiation of the base learner that is updated during the timesteps indicated within the box. The blue boxes each represent a summary statistic for the subtree they cover. Red highlights indicate active models, which will be updated or reinitialized based on the current time.

## 4.2  PTW results in Switching MNIST

We apply PTW with SGD as the base learner to our Switching MNIST task. As can be seen over the first 100,000 timesteps in Figure 5a, PTW ($87.3 \pm 0.3\%$) improves on SGD's performance ($85.0 \pm 0.3\%$). Furthermore, Figure 5b shows how over the full million timesteps PTW ($87.4, \pm 0.2\%$) does not suffer the same collapse in performance. In fact, none of its 30 seeds experienced the catastrophic loss of plasticity that hit all SGD learners.

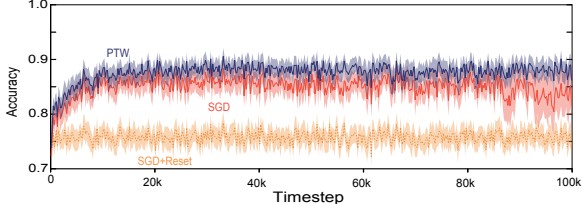

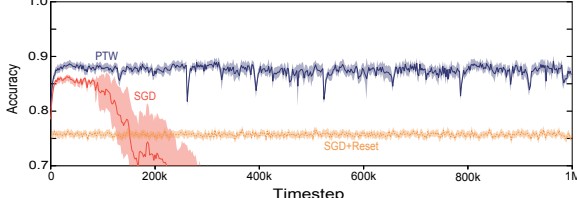

(a) Accuracy over the first 100,000 timesteps. Using a mixture of long- and short-term SGD models PTW outperforms the other experiential learner, SGD, and SGD+Reset.

(b) Long-term performance of PTW compared to SGD and SGD+Reset. While every SGD run stops tracking within a million timesteps, PTW maintains good performance across every seed.

Figure 5: Accuracy of PTW compared to SGD and SGD+Reset.

The binary reset points that PTW relies on for computational efficiency are a benefit and a curse. On the one hand, the binary resets minimize distance to the best partition with only $\lceil \log_2(t) \rceil$ models required. On the other hand, the reset times align across $i$ models for each multiple of $2^i$. When many models are reset at once, the overall cost in prediction accuracy is noticeable even in the heavily smoothed figures. This can be seen in Figure 5b, where drops in accuracy at regular power of 2 intervals are clearly visible. Furthermore, while the long-running models retain the cross-task generalization that SGD is able to discover, at reset that knowledge is lost. PTW mixes over long-running models with that generalization and short-running models without. Ideally we would be able to carry what is legitimately generalizable across reset boundaries. Then even when the long-running models fail to track, their representational power can still be shared with the newly-initialized models.

# 5   Using MAML with PTW

To improve the initial performance of short-running models, we bring Model-Agnostic Meta-Learning (MAML) into our toolbox (Finn et al., 2017). MAML was designed as an offline meta-learning procedure for discovering useful weight initializations for deep networks. Its meta-training defines the loss in terms of performance **after** some steps of adaptation, rather than the model's current performance. MAML training does this by embedding an update step in the meta-loss calculation, and calculating a change to the initial weights with respect to the updated model. This requires training data to be grouped by task, with carefully controlled sampling from distinct tasks. Before we look into adapting MAML for the experiential case, we will evaluate its performance offline and with task-boundary notification, then show that the offline-trained MAML initialization improves both SGD and PTW's performance.

## 5.1   MAML Implementation

MAML is a gradient-descent learning algorithm with a meta-loss function split into an inner and outer loop. The inner loop executes task-specific tuning of the network weights, using the gradient of the loss on a sample from the current task[9]. With another sample from the same task, the outer loop then uses the post-adaptation loss (*i.e.*, the gradient of the loss with respect to the fine-tuned weights) to adjust the pre-adaptation weights. This training process repeats offline on randomly sampled tasks, resulting in a set of network weights ($\text{MAML}_{\text{init}}$) that provide fast adaptation for few-shot learning (Finn et al., 2017). There are many variations on this simple training regime, but we use a simple setup and allow hyperparameter tuning for the highest average accuracy on the experiential task.

Although it is defined as an experiential task, we can exploit the piecewise-stationary structure of Switching-MNIST to construct an offline training regime. For every training step, we sample a random task and corresponding training batch, then split it into a *tuning batch* for the inner loop and *validation batch* for the outer. We chose to split the data evenly between the update and validation batch. This split was the most consistent: other divisions showed only marginal improvement in some switching regimes, and were less robust overall. Experiments with different training batch sizes did not show any consistent improvement in results, so we used a training batch size of 16 so that the tuning batch matched the online batch used in the experiential setting.

The MAML model has a standard construction with cross-entropy loss, a simple SGD update rule in the inner loop and Adam optimiser in the outer loop. The meta-loss function uses the tuning batch to execute a single gradient step from the initial network weights, and then calculates the cross-entropy error of the updated weights on the validation batch. For offline MAML training, we sample from 100,000 random tasks, which we found had the best online performance without risk of overtraining. As others have noted, MAML can be sensitive to hyperparameter settings and we found a larger training set greatly increased the variance across seeds without significantly improving performance (Antoniou et al., 2019; Nichol et al., 2018).

For the actual experiment, the pretrained weights ($\text{MAML}_{\text{init}}$) are used by two SGD learning algorithm as in Section 3.2. In the task-agnostic case, After initializing the network with $\text{MAML}_{\text{init}}$, MAML+SGD uses SGD to fine-tune continually: *i.e.*, without resetting for the duration of the experiment. MAML+SGD+Reset uses both MAML pretraining and task-aware resetting. At every task switch the SGD network weights are reinitialized to $\text{MAML}_{\text{init}}$. As before, performance is evaluated with per-timestep average accuracy, over the same 30 environment seeds.

## 5.2   MAML results in Switching MNIST

The initial weights MAML learns provide a clear benefit for a non-experiential SGD learner, as shown in Figure 6. Not only does MAML+SGD+Reset ($96.5 \pm 0.6\%$) achieve much higher average accuracy than SGD+Reset ($75.8 \pm 0.1\%$), it also outperforms SGD ($85.0 \pm 0.3\%$). On the other hand, when not explicitly reset at task boundaries, MAML+SGD ($83.6 \pm 1.7\%$) only briefly benefits over random initialization.

---

[9]Or tasks.

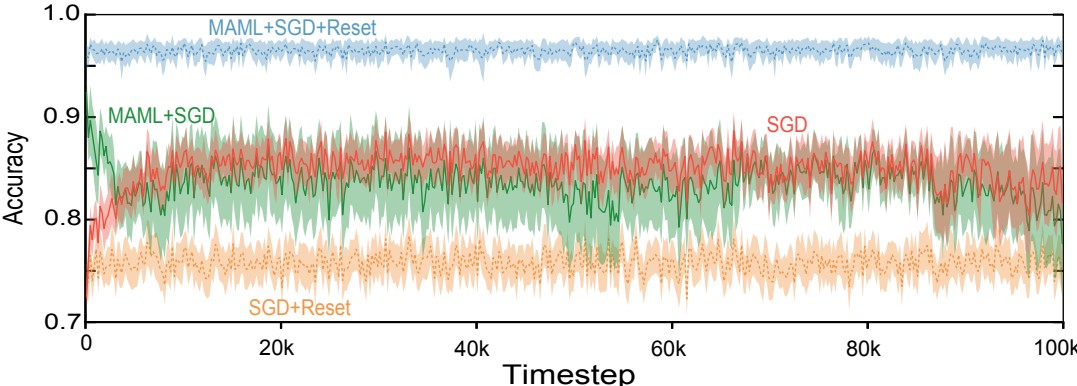

Figure 6: Effect of MAML pretraining compared to random initialization in the first 100,000 timesteps. The offline-trained, task-aware MAML+SGD+Reset significantly outperforms the more experiential learners. Although the MAML pretraining provides 100,000 offline updates, MAML+SGD only briefly outperforms SGD and has higher variance.

The benefits of starting from a MAML initialization are clear, but as it requires offline training and knowledge of task boundaries, it does not apply to the experiential task setting. The next step is to test MAML pretraining with PTW for continual learning, then explore relaxing the requirement for offline training.

## 5.3 PTW results in Switching MNIST

The combination of MAML and PTW for learning without task boundaries is a simple matter of pretraining a MAML initialization (MAML$_{init}$) and providing it to a PTW learner to use in place of random initialization. We will use MAML+PTW to refer to this combo. This shift from random initialization affects the PTW algorithm in two ways. First, the pre-trained initialization is immediately more accurate than a random initialization (this can be seen in the initial performance gap between MAML+SGD and SGD in Figure 2). Second, unlike PTW with random-initialization, all learners reinitialized on a given timestep will have identical network weights. This removes the ensemble-like benefit from mixing over multiple random initializations. For the offline MAML+PTW approach to succeed, the benefit from MAML's initialization will have to outweigh the loss of the ensemble.

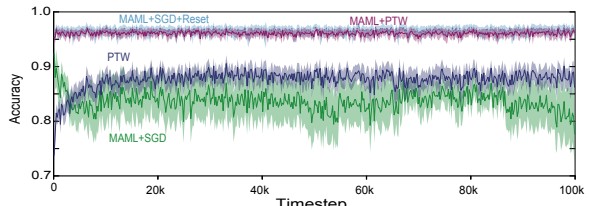

(a) MAML+PTW performs almost as well as MAML+SGD+Reset over 100k steps without receiving side information about task boundaries. It also maintains that performance, where MAML+SGD

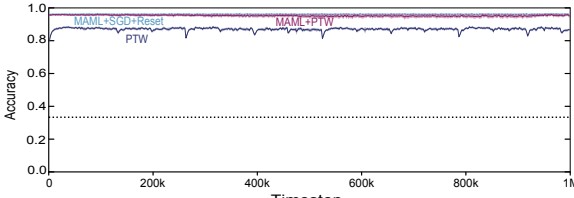

(b) Effect of combining PTW and MAML over one million timesteps. MAML+PTW is both more accurate and more consistent than PTW, and does not experience the same drops in accuracy as PTW at binary timesteps.

Figure 7: Effect of combining PTW and MAML.

As we hoped, MAML provides a dramatic improvement in PTW's performance, as seen in Figure 7b. The average accuracy of MAML+PTW over one million timesteps ($96.1 \pm 1.5\%$) is significantly greater than that of PTW ($87.4, \pm 0.2\%$). The pretraining provides an immediate and lasting benefit: initial accuracy is higher and the cost of resetting is now mitigated by the better initialization. The absence of the sharp drop in

accuracy at timestep $2^{18} \approx 262k$, for example, shows that the MAML initialization used by MAML+PTW allows the learners to recover more quickly than the random initializations used in PTW alone.

The addition of PTW lets MAML be used, through $\mathrm{MAML_{init}}$, for truly continual learning, albeit with offline training. Over 100,000 timesteps, as seen in Figure 7a, MAML+PTW ($96.2 \pm 0.5\%$) is significantly better than both PTW ($87.3 \pm 0.3\%$) and MAML+SGD ($83.6 \pm 1.7\%$), and almost as good[10] as the non-continual MAML+SGD+Reset ($96.5 \pm 0.6\%$). This is in spite of the fact that MAML+PTW incorporates no knowledge of task boundaries.

Though we did not set out to develop a continual MAML, the combination of MAML and PTW does create just that. The addition of PTW, in the form of the more advanced Forget-me-not Process, provided a similar benefit to the Elastic Weight Consolidation algorithm, allowing it to work in a task-agnostic setting (Kirkpatrick et al., 2017). As While the C-MAML proposed in Caccia et al. (2020) automatically detects task boundaries, and the OSAKA framework proposed there shares many similarities with our experiential learning framework, the learner still relies on offline pre-training and explicit task detection. FOML (Fully-Online Meta-Learning) and La-MAML (Look-ahead MAML) require that the tasks have no interference and focus on minimizing forgetting rather than long-term adaptibility (Rajasegaran et al., 2022; Gupta et al., 2020). All of these methods, including MAML+PTW, rely on offline training or explicit task recurrence. To use MAML for the fully experiential case, we must train it online as well.

## 6 Replay-MAML and PTW

Because we have PTW to take care of the online application of $\mathrm{MAML_{init}}$, we only need to adjust MAML's training process. For this we propose a simple MAML extension that uses a relatively small replay buffer and is trained entirely online. On every timestep we execute a standard MAML update using a random contiguous block from the circular replay buffer. Our tests show this works in our experiential setting even with a small buffer, a single gradient step per timestep, and training batches that are in no way forced to align with task boundaries.

### 6.1 Replay-MAML Implementation

The Replay-MAML buffer is a circular buffer where each index contains a single $(\mathbf{X}, \mathrm{y})$ pair. At each timestep, the input-class pairs from the current timestep are written into contiguous indices in the buffer. When the write index reaches the end of the buffer, it is moved back to the start so that new samples overwrite the oldest. This simple structure has several important benefits: the meta-training does not have to use the same batch size as the online experience, and the meta-learned weights are updated on many different tasks even though the buffer may be small and holds only the most recent tasks. However it also means that when sampling from the buffer, Replay-MAML provides no guarantees that the sample is all drawn from the same distribution: it is entirely possible that the training sample straddles one or more task changes. In our tests, discussed below, this proved to have surprisingly little effect on performance.

The online meta-training procedure is simple: randomly select an index from the buffer, take the meta-batch of samples, shuffle them, and split them into update and validation sets for the normal MAML update. Note that regardless of the total size of the buffer, the first meta-training update happens as soon as the buffer holds enough samples for a single training batch. Thereafter Replay-MAML does one meta-training update for each timestep.

For online prediction, the meta-trained weights can be used directly in place of the offline-trained $\mathrm{MAML_{init}}$. Whenever the online learner needs to initialize network weights, it queries the Replay-MAML component and uses the most recent values[11]. Thus on the first few timesteps, the Replay-MAML weights are no worse than random initialization, and this starting point quickly improves.

---

[10] Although the confidence intervals overlap due to the noisy task, MAML+SGD+Reset's per-timestep accuracy is consistently slightly above MAML+PTW.

[11] For fairness we ensure that the learner does not have access to the updated meta-trained weights until after its performance is evaluated.

## 6.2 Replay-MAML performance

The experiments shown here used a meta-training batch size of 16 and split the data evenly between inner and outer loss calculations. Our default buffer size is 1,250 timesteps. See Appendix A.1 for experiments varying the buffer size.

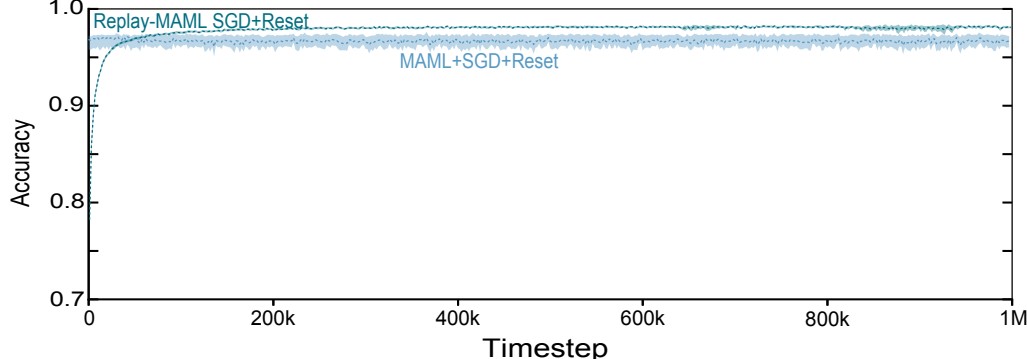

Figure 8: Replay-MAML compared to MAML+SGD+Reset over one million timesteps. The online SGD+Reset quickly reaches and then surpasses the performance of the offline MAML+SGD+Reset. Though online, these are not continual learners as they are explicitly reset at task boundaries.

As can be seen in Figure 8, Replay-MAML SGD+Reset quickly improves its random initialization. Within 25,000 timesteps it has reached MAML+SGD+Reset's average accuracy $(96.5 \pm 0.6\%)$, though it has been updated a fraction of the time with samples from 500 different tasks rather than 100,000. By 100,000 timesteps, the online Replay-MAML SGD+Reset $(97.8 \pm 0.0\%)$ outperforms the offline MAML+SGD+Reset $(96.5 \pm 0.6\%)$[12]. In spite of sampling from a relatively small replay buffer and not respecting task boundaries, Replay-MAML SGD+Reset produces a successful MAML initialization, completely online.

## 6.3 Replay-MAML PTW

Finally we have all the pieces we need for our fully experiential learner—no pretraining or knowledge of tasks and task boundaries required. It inherets only the constraints of the component algorithms (incremental for efficient PTW, and differentiable for MAML) and scales in $\lceil \log_2(t) \rceil$ rather than number of tasks. We call this combination Replay-MAML PTW.

As before, we run Replay MAML and PTW in parallel from random initialization. Where Replay-MAML SGD+Reset required explicit notification of task changes, Replay-MAML PTW relies on PTW for resets. Where MAML+PTW required pretraining and PTW allows for no cross-reset generalization, Replay-MAML PTW resets its learners by querying the Replay-MAML meta-training component for the most recent initialization weights.

The performance of Replay-MAML PTW is shown in Figure **??**[13]. The beneficial effects of MAML initialization hold even when trained online: Replay-MAML PTW surpasses the accuracy of both PTW $(87.3 \pm 0.3\%)$ and SGD $(85.0 \pm 0.3\%)$ within 5,000 timesteps. Figure 10 contrasts Replay-MAML PTW with MAML+PTW, in finer detail over one million timesteps. Here Replay-MAML PTW $(96.4 \pm 1.1\%)$ shows the the same stability we saw in MAML+PTW $(96.1 \pm 1.5\%)$, with consistent performance over one million timesteps. The fully experiential Replay-MAML PTW is able to surpass the performance of MAML+PTW because meta-

---

[12]We found it surprising that Replay-MAML could surpass offline MAML. One reason appears to be that training online naturally gives rise to repeated samples from the same task. Rather than being a problem, especially for long-horizon tasks, it appears to be a benefit. Recall that our offline MAML training procedure samples one batch per task. Various modifications of the offline MAML training allow for more batches per task, but we could not find a parameter setting that remained stable in the long-running task. As parameter tuning for MAML is known to be tricky (Antoniou et al., 2019; Nichol et al., 2018) and we are not making claims about state-of-the-art performance, we chose to stick with the stable hyperparameters for our results.

[13]Results are shown for the best-performing hyperparameters, averaged over 30 trials, as before.

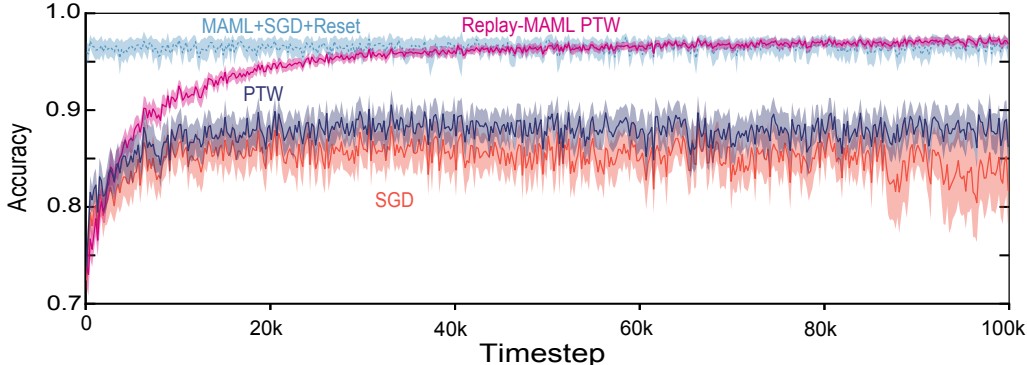

Figure 9: Replay-MAML PTW performance compared to the other experiential learners and MAML+SGD+Reset. Within the first 5,000 timesteps Replay-MAML PTW surpasses both PTW and SGD. Furthermore, within 100,000 timesteps it matches the performance of MAML+SGD+Reset.

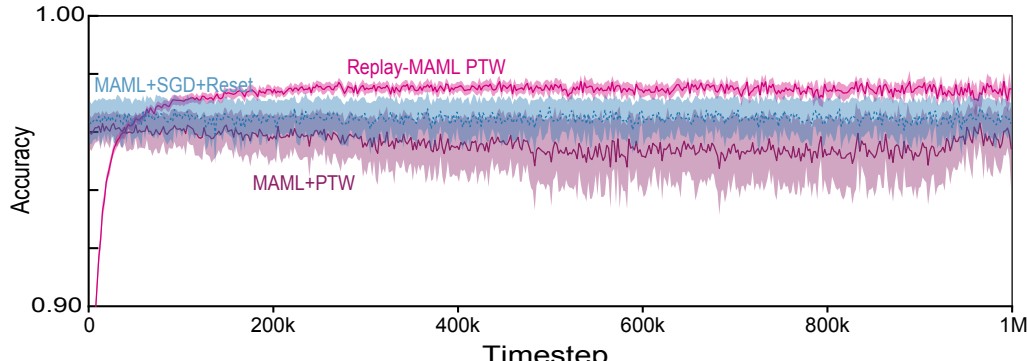

Figure 10: A longer running, more detailed look at Replay-MAML PTW's performance. It quickly surpases MAML+PTW, showing the online training creates a superior MAML$_{init}$. After the first 100,000 steps Replay-MAML PTW surpasses even MAML+SGD+Reset.

training online allows the experiential learner to continue to adapt, while MAML+PTW is forever bound by the performance of the offline trained model. Furthermore, Replay-MAML PTW improves on the performance of the offline, task-aware MAML+SGD+Reset ($96.5 \pm 0.6\%$). In this setting, Replay-MAML PTW is able to improve on the best pre-trained, task-aware learner while operating entirely within our experiential constraints.

# 7 Conclusion

In this paper we introduced Replay-MAML PTW, an approach to experiential learning that combines two meta-learning techniques: PTW for online learning in non-stationary settings, and MAML for learning to initialize a model that adapts quickly. We demonstrated our approach in a piecewise-stationary classification task. We showed that not only does it mitigate the eventual collapse of traditional gradient descent methods, it is able to continually adapt, even outperforming methods that make use of offline pretraining and are informed explicitly of the non-stationary task changes. The benefits of Replay-MAML PTW does come at a $O(\log_2 t)$ computation and memory cost. In the future, we hope to adapt our approach to the continual reinforcement learning setting, and explore reducing the additional small computation cost by restricting PTW's restarts to subsets of the networks.

**Broader Impact Statement**

One reason most machine learning systems focus on producing a static classifier, policy, or model is that analyzing the performance of static systems is much easier than analyzing the performance of a dynamic process. Introducing powerful techniques for continual learning may lead to hasty deployment in real-world applications, and we don't yet have the legislative, regulatory, or post-deployment support structure to handle systems that dynamically respond to their specific context and that change in response to unique histories. Development of applications with continual learning should proceed cautiously and in concert with development of the ethical and practical framework for handling their deployment. Luckily, we have time. Continual learning research is in very early stages.

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

## A  Effect of buffer size on Replay-MAML

### A.1  Replay-MAML results in Switching MNIST

Combining Replay-MAML with another learner thus introduces a computational cost on each timestep for the MAML update, and a memory cost for the replay buffer to store previously observed samples and tasks. The computational cost of Replay-MAML is roughly equal to the cost of an update for SGD+Reset, which we consider an acceptable tradeoff for eliminating the need for MAML pretraining and *a priori* knowledge of the task distribution that will be encountered during the online phase. The memory cost of Replay-MAML depends on the size of the replay buffer used. A large replay buffer is expensive, but random samples drawn from it will more closely approximate the stream of new tasks used for training MAML. A small replay buffer is cheap, but random samples from it will be more likely to repeat the most recently observed tasks, or even the current task. We will investigate this tradeoff empirically, to discover how small the replay buffer can get while still providing good performance.

As can be seen in Figure 11, Replay-MAML quickly improves its random initialization, reaching and even exceeding offline MAML's average accuracy of $96.5 \pm 0.6\%$ (indicated on the graph with the dotted line). The red line illustrates our default buffer size. Note that within 2500 update steps, having seen on average only 50 different tasks, it is performing as well as MAML trained on 100,000 independent samples. In fact, we see that while larger buffers improve the early performance of the learner, all four of the learners match

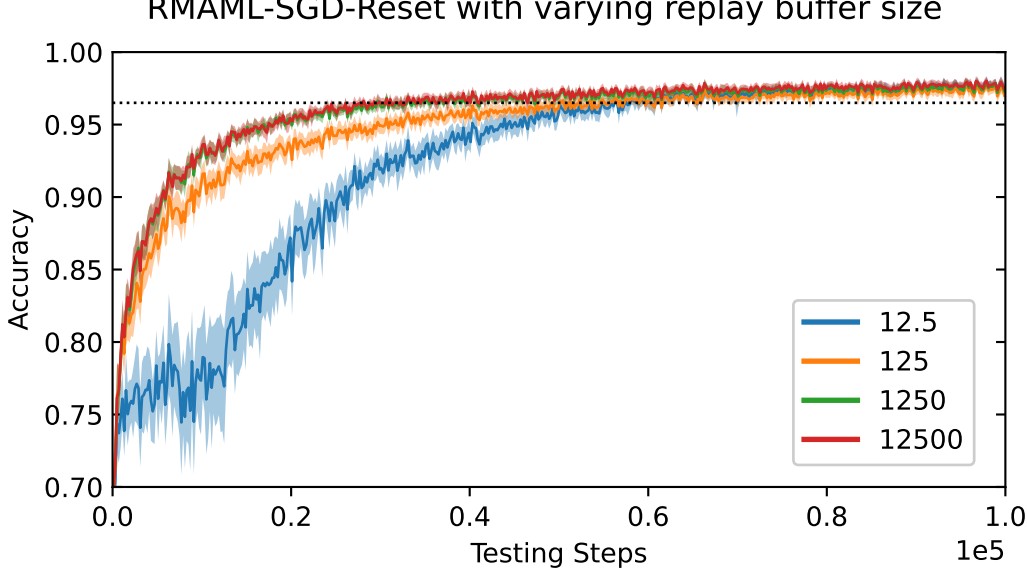

Figure 11: Average online accuracy of SGD+Reset trained with four replay buffer sizes, averaged across 30 random seeds. The legend indicates the timesteps spanned by the buffer.

and then slightly surpass the accuracy of MAML+SGD+Reset, while training on about half of the 100,000 batches used for MAML pretraining. Further, relatively small replay buffers are sufficient: the 1250 buffer size learner stores just 1.25% of the data observed online, with no loss in accuracy compared to the 12500 buffer size learner.

## A.2 Online-MAML results in Switching MNIST

We found Replay-MAML's resilience to small buffer sizes surprising. We had expected when the buffer was smaller than the average task length, Replay-MAML's performance would suffer dramatically. And although larger buffer sizes perform better, the fact that meta-training is executed almost exclusively on the current data distribution does not stop Replay-MAML from learning a very good initialization. Naturally, we wondered if we could do away with the buffer entirely.

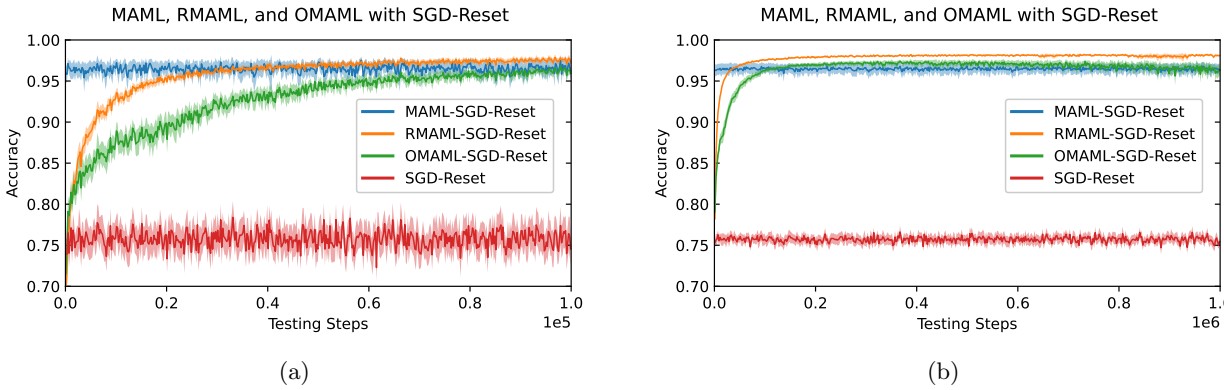

Figure 12: Average online accuracy of 30 independent runs of offline (MAML), replay-buffer (Replay-MAML), and online (OMAML) MAML learners.

With no replay buffer, our online MAML update (OMAML) is simply that: execute the meta-training MAML update as always but online, using only the current batch of data. For simplicity we simply divided

each batch in half, leaving us with 4 samples for the inner SGD step and 4 for outer meta-update. When the environment signals that a task has been reset, the online SGD+Reset learner pulls the most recent parameter set from Replay-MAML[14] and reinitializes to that.

The performance of OMAML is illustrated in Figure 12. The average performance of the best offline-trained MAML-SGD+Reset is shown in blue. Although OMAML does not exceed offline MAML's average performance within the first 100,000 updates, it is able to match it in the long run.

---

[14]Note for fair comparison we do not allow the Replay-MAML weights to be updated by the current data until the accuracy is evaluated

