# OpenReview forum: "Experiential Learning: Using Partition-Tree Weighting and MAML in the Continual and Online Setting"
_TMLR — Rejected by TMLR_

### Review · Reviewer_a3dD · 2023-08-30

**Summary Of Contributions:**

The authors propose a solution to the problem they call "experiential learning", which seems to refer to the subset of continual learning where evaluation is on-line and no task IDs or boundaries are provided to the learner.


IIUC, their solution consists in maintaining a certain number of learners, reinitialized at different intervals (every 1, 2, 4... 2^D steps), and use MAML (based on a relatively short-horizon replay buffer arranged as a queue) to optimize the parameters used for reinitialization.

They show that the method performs well on a specific problem (the so-called "switching MNIST" problem, introduced by the authors) where the classes and their labels change at unpredictable (and unannounced) times.

**Audience:**

Yes

**Broader Impact Concerns:**

There does not seem to be any broader impact concerns.

**Claims And Evidence:**

Yes

**Requested Changes:**

*Major:*

The method is not fully described. Section 4.1 tells us that the method maintains multiple learners, reinitialized at different intervals, and then... what? How are these learners used to produce a single answer for each new item? What is the "cost" mentioned at the end of p. 6? What are the "summary statistics", and how are they used?

Also, which value of D is actually used in the experiments?

Evidently, a large chunk of the method description is missing.


*Debatable:*

The method is only evaluated on one novel domain: the so-called "switching-MNIST" introduced by the authors, which seems to be essentially a smaller version of the Omniglot task based on MNIST digits (and where classes include multiple underlying characters). No other task from the literature is used. OTOH, the task  is certainly a continual learning task, and satisfies the authors' requirements for experiential learning. Whether it is sufficient to demonstrate the applicability of the method, this is a choice I leave to the editors.


*Minor:*

The paper feels generally rushed, with multiple typos and nonsense sentences. Here is a sample:

- Section 2: "evaluation still be understood" - missing word?
- p. 3: "from the what Sutton", then "followed by abrupt) to"
- Top of p.4: explain what k and n are! (number of classes and number of digits per classes, IIUC)
- top of p.5: "tasks may unlikely"
- 5.2, last paragraph: "algorithm"->"algorithms", "After"->"after".
- p. 10, 3rd paragraph: "As While"
- Section 6.3, 1st paragraph: "inherets"

**Strengths And Weaknesses:**

Strengths:
The problem is relevant, the method is novel and astute, and it seems to work.

Weaknesses:
The paper is incomplete. The method is not fully described, making replication impossible. Also, the evaluation may or may not be adequate.

---

### Review · Reviewer_fFgf · 2023-09-19

**Summary Of Contributions:**

The paper studies a learning setting where data are sequential and non-stationary, and the training and evaluation are performed online (called by the Authors “experimental learning”). The paper proposes an algorithm that combines Partition-Tree Weighting (PTW), Model-Agnostic Meta-Learning (MAML), and replay buffer. The method is evaluated on a piecewise stationary MNIST-based classification task (Switching MNIST). The paper provides empirical evidence that the proposed solution performs favorably against chosen baselines.

**Audience:**

Yes

**Claims And Evidence:**

No

**Requested Changes:**

See the Strengths And Weaknesses section.

**Strengths And Weaknesses:**

Strengths:
* Merging MAML and PTW is interesting.
* The proposed algorithm performs favorably against chosen baselines.
* The paper has several interesting observations (e.g., initialization effect trumps the ensemble effect (Section 5.3), learning on a small buffer with batches of data spanning more than one task does not hinder learning (Section 6.1)).



Weaknesses [General]:
- The paper should clearly and concisely describe goals and main contributions. It should gather the main results and statements scattered across the paper in one easy-to-access place (e.g., the last paragraph of section 2.2, the second and fifth paragraphs on page 10, etc.)
- Plan of experiments:
	- The paper considers multiple setups (e.g., a combination of various methods, online/offline, task boundary/no task boundary, resets/no resets). It is easy to lose track without a proper experiment roadmap. Consequently, the paper would benefit from clearly laying out the experiments', and highlighting the main results.
	- It would help if the paper included a table with the considered approaches and their corresponding setup. This could help flesh out all the baselines used in the paper and what are the differences between the models.
	- It would help if the experimental section names were informative with respect to the presented results. For instance, what combination of methods is used, is the training online or offline, is the task boundary available, etc.
- Presentation of results:
	- The scales on the Figures vary (e.g., Fig. 7a and 7b). It would help if a reader were informed about this, as it might cause unintended confusion.
	- It should be more clearly explained why some figures have 100k timesteps while others have 1M. In particular, since, in most cases, the learning curves stay on a similar level.
	- Some figures include information about the number of seeds, while some do not.
	- In Section 6.2, the approach Replay-MAML SGD+Reset is reported to be almost deterministic (+/-0%). What is the reason for that?
	- Figures differ in the choice of presented baselines. What is the reason for that?
	- In Section 5.1, it is mentioned that offline MAML training used 100,000 random tasks. How is this related to 100k timesteps or the fact that there are 18.9k possible different tasks?
- Choosing various combinations of components, the paper defines a couple of baselines. These do not include popular CL methods or architectures (such as packnet or progressive neural networks),
- The text includes multiple statements of a general nature without backing them up with appropriate citations. Some examples include: “truly robust and general machine learning systems incorporate continual adaptation”, “assumption underlying many machine learning techniques is that data is [...] (i.i.d.)”, “many continual learning techniques rely on pretraining with a separate, offline data source“, “sequential data is the only data there is, “continual learning is often synonymous with the study of catastrophic forgetting”, “[...] rarely investigate performance during training [...]“.
- Some interesting information is hidden in the footnotes (e.g., 5, 6, 11, 12). The Authors should consider putting this in the main text.
- Some typos include
	- “[...] by abrupt) to [...]” page 3.
	- “In the task-agnostic case, After [...]” page 8.
	- “Fig ??”, page 11.

Weaknesses [Text]:
- Section 1. Introduction:
	- The text should be more concrete and with clearly defined problems and contributions.
	- The feel of the text could suggest that “experimental learning” is a new concept, which seems unconvincing and should be better conceptualized (see also the remarks concerning the Related Work section).
- Section 2. Background:
	- Experimental learning
		- From the context, it is unclear what the difference is between “experimental learning” and continual learning with no task boundary.
		- It needs to be clarified from the context how this concept differs from open-endedness.
		- The three keywords used in the definition (“embedded”, “immediate”, and “ongoing”) loosely relate to their corresponding description.
	- Related work misses some important lines of research, including open-endedness [1] or environment design [2].
	- Some statements are phrased in a manner that could mislead an unaware reader without giving a proper citation. “continual learning is often synonymous with the study of catastrophic forgetting”, “[...] rarely investigate performance during training [...]“). Even if some papers fall under this category, the related work should focus on the research that addresses problems described by the Authors.
	- The treatment of reinforcement learning needs to be deeper. In particular, more attention needs to be paid to RL papers that study and suggest solutions to similar problems to the ones considered in this paper. The learning process in RL is non-stationary due to off-policy learning, changing policy, changing auxiliary rewards, etc.
	- There is little mention of continual RL literature.
	- Some text fragments could be moved to this section, e.g., the last paragraph of Section 5.3.
- Section 3. Experiments:
	- Switching MNIST:
		- The task is rather toyish, which puts the paper on the proof-of-concept side.. The text should recognize this.
		- The paper should make the case that the task is interesting enough to be studied (e.g., that it approximates some bigger tasks of interest).
		- The exposition should be improved, including the main objective of the task, how the task is evaluated and what metrics are used, a clear step-by-step explanation (possibly with a pseudo-code), the definition of $n$ and $k$ with their default values, the switching mechanisms with its default switch probability value, and the objective of the task.
		- It would also be nice to see some discussion about the label inference problem and what challenges it brings to the table.
- Section 4. PTW:
	- PTW, one of the method's main components, should be clearly explained. For example, are the ensemble elements added online or initialized at the start, at what timesteps are models initialized, at what timesteps are models updated, how does the summary work, and how is the ensemble used? Some of the more technical details could be placed in the Appendix.
- Section 5.  MAML
	- The MAML could be shortened and made more precise with the help of several key formulas.

Some other questions:
- The method increases the cost of learning and inference (regarding memory or wall time). How would the approach scale to tasks requiring large neural networks?
- If the effect of initialization dominates the ensemble effect, can the approach be reduced to a single network with some form of resets and reinitializations?
- The approach emphasizes the continual learning process. However, the learning curves show that the agents learn fast to solve the problem, and nothing else is happening (except maybe for collapse). This raises questions concerning the task choice or the approach.


[1] Kenneth Stanley, Joel Lehman, and Lisa Soros. Open-endedness: The last grand challenge you’ve never heard of. 2017.

[2] Michael Dennis, Natasha Jaques, Eugene Vinitsky, Alexandre Bayen, Stuart Russell, Andrew Critch, and Sergey Levine. Emergent complexity and zero-shot transfer via unsupervised environment design. NeurIPS 2020.

---

### Review · Reviewer_XN27 · 2023-10-02

**Summary Of Contributions:**

The paper introduces the concept of "experiential learning," an extension of continual and online learning paradigms. This concept is based on the ongoing and simultaneous flow of data, evaluation, and learning, addressing non-stationary data distributions. To implement this, the paper merges two pre-existing algorithms, Model-Agnostic Meta-Learning (MAML) and Partition-Tree Weighting (PTW), forming a new algorithm termed Replay-MAML PTW. This algorithm is designed to continually adjust and learn from sequential data, mitigating the need for distinct re-training phases. The approach is exemplified using a newly designed task, "Switching MNIST," derived from the MNIST dataset, aiming to simulate experiential learning environments. Through meticulous experimentation, the paper isolates and evaluates each element of the proposed method to understand its impact.

**Audience:**

Yes

**Broader Impact Concerns:**

N/A.

**Claims And Evidence:**

Yes

**Requested Changes:**

__Critical__:

- Please address the issue of generalizability by potentially including results from other datasets like CIFAR-10, to strengthen the validity of the findings.
- Please consider comparing with additional and varied baselines to create a more comprehensive evaluation. See "candidate baselines" below.

__Strengthening__:

- Reconsider the need for coining the term ‘experiential learning’ and instead more explicitly define the specific focus area within the realm of continual learning, enhancing the precision and relevance of the paper’s conceptual framework.
- The paper should provide more detailed and formal descriptions of MAML and PTW, potentially in an appendix if space in the main text is a constraint, to facilitate better comprehension of these core building blocks.

__Questions__:
- Regarding Figure 7(b), what is the performance level of MAML+SGD? Is there an observed loss of plasticity within this method as well? Understanding this could elucidate whether the advantageous initialization acquired through offline MAML training can inherently circumvent issues related to plasticity loss.
- In scenarios where there is substantial interference between tasks, is the efficacy of Replay-MAML maintained?

[Candidate baselines]:

A. Pre-train then fine-tune. Unlike Permuted MNIST, the Switching MNIST task retains the semantics of images. Hence, during the pre-training stage, the representations shared by all tasks and images can be learned, which can aid in online fine-tuning afterward.

B. Ensembled SGD. For a fair comparison with PTW which maintains $log_2 t$ models, you can train an ensemble model with SGD with $log_2 t$ models at max.

C. PTW + partial reset. Instead of resetting the whole parameters of a model, one can partially re-initialize some parameters. I am curious whether this will lead to a better performance than simply doing PTW + reset.

**Strengths And Weaknesses:**

__Strengths__:

- The paper offers well-designed experiments that shed light on the individual impacts of different components of the proposed algorithm, providing nuanced insights into its functionality.
- The proposed idea is cogently presented and comprehensible, offering a straight-forward approach.
- The inception of a novel task, Switching MNIST, serves as a meaningful and practical testbed.

__Weaknesses__:

- The paper sometimes falls short in clarity, especially regarding the presentation of core concepts like MAML and PTW. A more thorough and formal definition of these concepts is needed for a more holistic understanding.
- The necessity to coin a new term, 'experiential learning', is debatable. A clear definition and focus on the niche area the paper is addressing (within continual learning) could have been more beneficial than introducing a new term.
- The evaluations predominantly rest on the Switching MNIST task, raising questions about the generalizability of the findings across different domains.
- The paper could have included comparisons with stronger and more varied baselines and considered alternate approaches like ensembled SGD and partial resets, offering a broader spectrum of insights (more on this below).

---

### Review · Reviewer_cxu5 · 2023-10-09

**Summary Of Contributions:**

The paper proposes to tackle the problem of " experiential learning" - an extension to online continual learning with a method formulated as the combination of the Model-Agnostic Meta-Learning (MAML) and Partition-Tree Weighting (PTW). The proposed method, termed as Replay-MAML PTW is demonstrated through experimental observations to outperform task-aware learners with an offline pre-training budget. For the purpose of experimentation, to satisfy the constraints, the paper proposes a new dataset - Switching MNIST. The paper goes on to demonstrate the effectiveness and viability of each component of the proposed method on the Switching MNIST benchmark, namely, 1. Replay; 2. MAML; and 3. PTW.

**Audience:**

Yes

**Claims And Evidence:**

Yes

**Requested Changes:**

1. Encouraging the authors to add more details on the components used in the proposed algorithm - MAML and PTW. For example, it is unclear from the text as written in Fig 6 that MAML+SGD has higher variance while briefly outperforming SGD at the start - The reason on why this might be the case is not obvious and could be important to be well understood by providing more details on the MAML algorithm.
2. Use of more baselines and constraints: The authors are encouraged to consider using more baselines - stronger datasets and models to demonstrate the efficacy of the proposed algorithm.
Further, the authors are encouraged to experiment with more accurately defined real-world constraints like that of compute where past works [3] have demonstrated that several common CL algorithms fail compared to that of vanilla baselines that samples uniformly from memory.
3. Authors are encouraged to revise the manuscript by correcting any outstanding grammatical errors and incompleteness in statements.

[3] Prabhu, Ameya, Hasan Abed Al Kader Hammoud, Puneet K. Dokania, Philip HS Torr, Ser-Nam Lim, Bernard Ghanem, and Adel Bibi. "Computationally Budgeted Continual Learning: What Does Matter?." In Proceedings of the IEEE/CVF Conference on Computer Vision and Pattern Recognition, pp. 3698-3707. 2023.

**Strengths And Weaknesses:**

Strengths:
1. The paper is easy to follow and structurally well written.
2. The motivation for the problem is well justified and provides intuitive context to be of interest of the wider community.
3. The experimentation protocol used for defining Switching MNIST is fittingly formulated and the experiments with each individual component of the proposed algorithm - MAML + Replay + PTW to assess their impact shows due diligence in evaluation efforts.


Weaknesses:
1. $\textbf{Definition of Experiential Learning}$: The papers point out that the evaluation on a static dataset as is the case in most continual learning (CL) benchmarks doesn't satisfy the constraints defining "experiential learning". However, it is my understanding that such evaluation is purely from an experimental perspective and doesn't limit CL to just static evaluation. Further, data arriving in a sequence with potential anytime evaluation on an environment with data drift are guiding principles for the sub-field of Anytime Learning (AL) [1][2]. Hence, "Experiential Learning" seems to be a byproduct of CL + AL and it is hard to argue if there is a huge distinction to warrant a new term being coined. Although, it agreeably improves readability and consistency, it adds more jargon to a very complex field.
2. Claim in Paragraph 2 of Section 2: "Many continual learning techniques rely on pretraining with a separate, offline data source." - This claim while may be evident in many CL works doesn't necessarily serve as a drawback. With the current availability of several large scale pre-trained models providing strong feature extraction capabilities, it remains to be understood from text as to why this might be of potential drawback. Rather it would be more beneficial to start from these pretrained models to improve upon the experiential learning task. This is not necessarily a weakness but rather a conflicting argument presented in the paper.
3. Coverage of experimentation: While proposing a new testbed - Switching MNIST for experimental purposes, the manuscript rather has a small coverage of experimentation - only relying on a small 3 layered network and with one dataset with MNIST images doesn't do justice to the proposed algorithm and the problem of "experiential learning".
4. The paper's lack of competing baselines - There are several CL algorithms which can be applicable and extended to be verified on this new testbed which should be reported.
5. The paper has some grammatical issues and incompleteness, for example, in Figure 7 (a) The caption abruptly ends at "MAML+SGD".



[1] Connie Loggia Ramsey and John J Grefenstette. Case-based anytime learning. In Case Based Reasoning: Papers from the 1994 Workshop, pages 91–95. AAAI Press Menlo Park, California, 1994.

[2] Lucas Caccia, Jing Xu, Myle Ott, Marc’Aurelio Ranzato, and Ludovic Denoyer. On anytime learning at macroscale. arXiv preprint arXiv:2106.09563, 2021.

---

### Author Response · Authors · 2023-10-16
**Thank you to all the reviewers**

Thank you to all the reviewers for their careful attention and thoughtful commentary. We are working on addressing the points.

The concerns about introducing yet another term into an overcrowded field are entirely valid, of course. We had found using either ``continual learning'' or ``online learning'' consistently resulted in confusion, in both the text and in reader's expectations, hence the development of ``experiential''. Thank you for the pointers to Anytime Learning, unsupervised environment design, and budgeted learning. The Anytime Learning at Macroscale paper has an excellent section situating Anytime Learning against online learning and other known frameworks, and the clarity this provides is inspirational. In addition to the frameworks mentioned by reviewers, we will contrast with online learning and specifically contextual bandits.

Apologies for the typos and other issues with the text. We have fixed the ones explicitly mentioned and will review our updated draft carefully once the major changes have been made.

Regarding the explanation of PTW and Replay MAML, we are extending the main text to make the procedure more clear. In addition, we will have full details in an Appendix for each approach, including how hyperparameters were chosen. We also intend to release the source code for ease of replication. We agree with the unanimous assessment that more detail is necessary for reproducibility.

The experimental domain is indeed a toy problem and not intended to represent the most realistic setting. We have run the same experiments with CIFAR10  and the mutating sinusoid regression task from the MAML paper, with the same results. We plan to run formal experiments on CIFAR10 and with Permuted MNIST, and update the text with those results. As Permuted MNIST creates a complete change in the input data while keeping the class labels consistent, in some sense, it should be an interesting contrast to the Switching approach. Omniglot, with only 20 images per character, does not lend itself to an infinitely-repeatable structure.

We are open to suggestions of other domains, especially where it is possible to run for long enough to see the catastrophic loss of plasticity. Reinforcement learning of course provides excellent fodder for this kind of research, but we felt there was enough complexity in writing up the supervised learning case to restrict ourselves for now.

The issue of what baselines to compare against was one we debated frequently, and as you can see in the end we went with the  minimal set. Given the unanimous discomfort with that, we are discussing the best way to extend the results with additional baselines, while still restricting to methods that can use the same network architecture. In particular, we are considering partial resets, an ensemble SGD, and a pretraining variant. In general, pretraining on the switching task simply hastens the catastrophic loss of plasticity.

---

### Decision · Action_Editor_cLHS · 2023-11-13

**Recommendation:** Reject

**Comment:**

The initial reviews all found the proposed work to be interesting. Reviewers agree that the setting is well-motivated, the work contains useful observations, and the method has elements of novelty (even though this is not part of TMLR's acceptance criteria). Nonetheless, the reviewers also found the current version of the paper to be somewhat preliminary for TMLR and made several suggestions for improving it.

The authors acknowledged the value of several of the reviewers' suggestions, but since these are akin to "major revisions" the consensus is that the best way forward is to reject this version of the paper and encourage the authors to re-submit an improved version once it's ready.

For the record, I summarize the elements that seem to be the most important (mostly reworded from the available reviews and the authors' reply):
+ Re-consider the introduction of the term "experiential learning."
+ Improve the general exposition of the paper, including its contributions, goals (several specific comments by reviewer fFgf), and the exposition of some of the background (Replay-MAML and PTW) and the main method (see comments from reviewer a3dD)
+ Extend the experimental studies (beyond Switching MNIST task) by considering additional domains, baselines (e.g., see specific suggestions by reviewer XN27), and architectures.

**Audience:**

The findings would be of interest to the TMLR community, in particular to the (supervised) continual learning and meta-learning audiences.

**Claims And Evidence:**

The reviewers made several suggestions regarding improving the clarity of the exposition, the experimental settings, and the results, including additional datasets and baselines. More details are in the "comment" field below.

**Resubmission Of Major Revision:**

The authors may consider submitting a major revision at a later time.